

# Multivariate statistical evaluation of dissolved heavy metals and a water quality assessment in the Lake Aha watershed, Southwest China

Shilin Gao[1], Zhuhong Wang[2], Qixin Wu[1,3] and Jie Zeng[4]

[1] Key Laboratory of Karst Geological Resources and Environment, Ministry of Education, Guizhou University, Guiyang, Guizhou Province, China
[2] School of Public Health, Key Laboratory of Environmental Pollution and Disease Monitoring of Ministry of Education, Guizhou Medical University, Guiyang, Guizhou Province, China
[3] The College of Resources and Environmental Engineering, Guizhou University, Guiyang, Guizhou Province, China
[4] Institute of Earth Sciences, China University of Geosciences (Beijing), Beijing, China

## ABSTRACT

Heavy metals are of public concern in aquatic ecosystems due to their growing release from industries and mining activities. This study investigated the sources, temporal-spatial distributions and water quality of dissolved heavy metals (Mn, Co, Al, Ni, Ba, V, Sb, Fe, Sr) in the Lake Aha watershed, an area under the influence of sewage and acid mining drainage. These heavy metals displayed significant spatial and temporal variabilities. The water quality index results (WQI values ranged from 3.21 to 15.64) and health risk assessment (all hazard indexes are below 1) indicated that dissolved heavy metals in this study pose a low risk for human health. Correlation analysis and principal component analysis indicated that Fe and Sr mainly presented a natural geological feature in the study area, and Mn, Co, Al and Ni were influenced by the acid coal mine drainage, whereas Ba, V and Sb were under the impact of local industrial or medical activities. This study provides new insights into the risk assessment of heavy metals in small watersheds.

## INTRODUCTION

With the growth of industrial, mining and economic activities, surface water quality is facing enormous challenges (*Islam et al., 2015*; *Ustaoğlu, Tepe & Aydin, 2020*; *Vörösmarty et al., 2010*). Among challenges, heavy metal contamination is of particular concern (*Ariffin et al., 2017*; *Burakov et al., 2018*; *Cameron, Mata & Riquelme, 2018*; *Islam et al., 2015*). Dissolved heavy metals in water could readily enter the food chain, which eventually leads to serious health risks to ecosystems and humans (*Nguyen et al., 2018*; *Rehman et al., 2018*). Heavy metals in water are primarily released from natural processes and anthropogenic activities (*Li & Zhang, 2010*; *Meng et al., 2016*). Natural processes include atmospheric dry/wet deposition, rock weathering, and volcanism, which are

Corresponding author
Zhuhong Wang, cindywzh@163.com

related to the local geology and lithology characteristics (*Han & Liu, 2004*; *Krishna, Satyanarayanan & Govil, 2009*; *Li & Zhang, 2010*; *Nriagu, 1989*). Anthropogenic activities including industrial, mining, medical and urban sewage can release a large number of heavy metals into the aquatic systems (*Han et al., 2019*; *Meng et al., 2016*; *Pekey, Karakaş & Bakogˇlu, 2004*).

To understand the health risks of heavy metals in waters, it is crucial to investigate the concentration, distribution and sources of heavy metals. According to the concentration, distribution, of heavy metals in aquatic systems, the sources of heavy metals from natural processes or human activities can be qualitatively identified via multivariate statistical approaches, such as correlation analysis and principal component analysis (PCA) (*Xiao et al., 2019*; *Zeng, Han & Yang, 2020*). Ingestion and dermal absorption are two main exposure pathways for aqueous heavy metals. To estimate the potential non-carcinogenic risks, the hazard quotient (HQ) has been commonly used in previous studies (*Ustaoğlu, Tepe & Taş, 2020*; *Wu et al., 2009*; *Xiao et al., 2019*), which is the ratio between exposure or average intake of contaminants and the corresponding reference dose (RfD). The water quality index (WQI) has been used to reflect the comprehensive influence of various dissolved heavy metals in water (*Meng et al., 2016*; *Taş et al., 2019*; *Ustaoğlu et al., 2020*; *Xiao et al., 2019*).

Lake Aha is a primary reservoir of tap water for Guiyang, the capital of Guizhou Province, SW China. With rapid economic development in the last decades, the ecological environment of the Lake Aha watershed is strongly impacted by urban and industrial activities. Several studies reported heavy metal contamination in the Lake Aha watershed, however, these studies mainly concerned heavy metals in solid phases such as suspended particulate matter and sediments (*Huang et al., 2009*; *Song et al., 2011*; *Zhang et al., 2019*). It becomes increasingly important to investigate the contamination, source and health risk of dissolved heavy metals in the Lake Aha watershed.

In this study, to facilitate drinking water management efficiency and provide a reference for the policy formulation of heavy metal pollution prevention, we systematically investigated nine heavy metals in three inflowing tributaries of Lake Aha. The aims are to (1) investigate the temporal-spatial distribution patterns of dissolved heavy metals in the tributaries; (2) identify the potential sources of heavy metals; and (3) assess the water quality and the ecological risk of dissolved heavy metals.

## MATERIALS AND METHODS

### Study site

The Lake Aha watershed is located in Guiyang City, Southwest China (26°34′N; 106°43′E), which holds a total area of 180.2 km$^2$ with a population of almost 150,000 and provides the vital water source for the local city (*Han, Tang & Xu, 2010*). The watershed is situated on the Yunnan–Kweichow Plateau, the global largest karst region, where karst aquifers and conduit networks are especially vulnerable to anthropogenic contaminations (*Sun et al., 2019*). The study site has a subtropical humid and monsoon climate, with a mean annual temperature of about 23.2 °C. The average annual rainfall ranges from 1,140 to 1,200 mm, while the average annual evaporation is 932 mm (*Bai et al., 2007*;

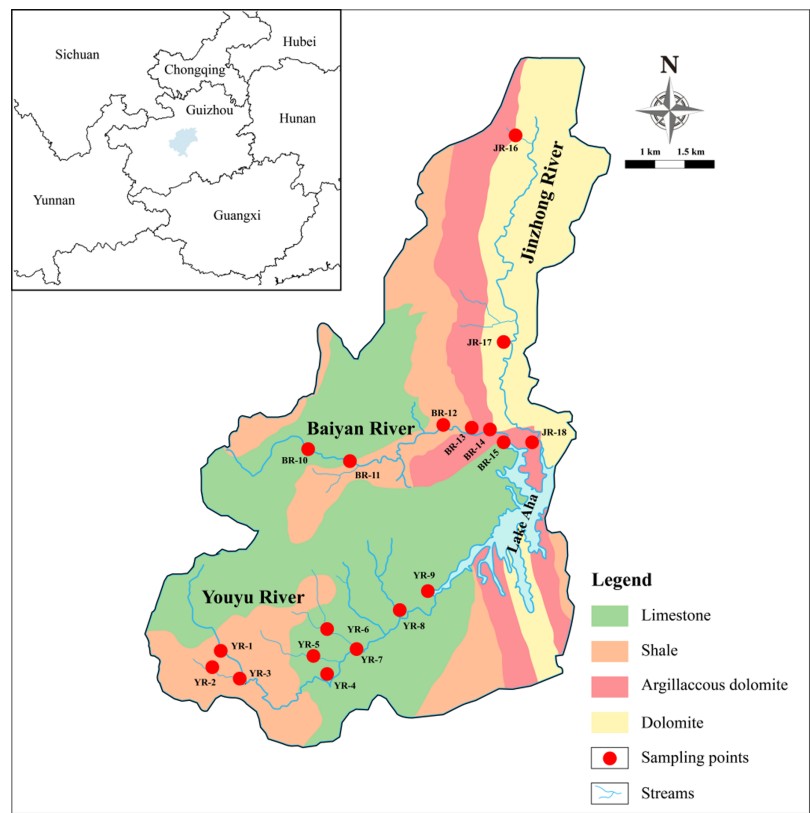

**Figure 1 Sampling sites in the tributaries of Lake Aha.**

*Pan et al., 2019*). The average annual rainfall days are about 178 days and most of the precipitation is mainly from May to August, accounting for about 65% of the total, with a relatively short annual average sunshine time over 1,412.6 h.

Lake Aha has three major inflowing tributaries, namely the Youyu River (YR), the Baiyan River (BR) and the Jinzhong River (JR). The bedrock of the YR is Permian limestone and Triassic shale, while that of the BR is Triassic dolomite and shale, and that of JR is Triassic dolomite (*Song et al., 2011*). The YR covers an area of 61.9 km$^2$ and flows eastward for 18.5 km through 11 villages. Agricultural land (53.0%) is the primary land use type of the YR watershed, whereas residential and commercial land is small (3.3%). The BR drains 51.5 km$^2$ and is approximately 15 km long. The primary land use type of the BR watershed is forest (40.2%). The JR has a length of 16.5 km and a watershed area of 47.5 km$^2$ with around 60,000 inhabitants. The dominant land use type of the JR watershed is residential and commercial land (81.6%), indicating the JR is intensively affected by human activities.

## Sampling and analysis

River water samples were collected from 10 cm below the surface water level, at 18 sites in the tributaries, once a month from November 2017 to April 2018 (low flow season). The sampling sites are shown in Fig. 1. Specifically, nine sites were located in the YR, six sites were located in the BR, and three sites were located in the JR.

Ultimately, a total of 103 river water samples were collected, however, five of them were damaged during transportation. The pH, electrical conductivity (EC) and dissolved oxygen (DO) values of water samples were immediately measured in the field using the WTW Multi3430 (WTW Company, Weilheim, Upper Bavaria, Germany). The samples were instantly filtered through 0.22 µm acetate cellulose-acetate membrane after collection. All the water samples were acidified with ultra-purified $HNO_3$ (pH < 2), then sealed in pre-cleaned polyethylene bottles and kept at 4 °C until determination.

## Heavy metals analysis

Nine heavy metals (Mn, Co, Al, Ni, Ba, V, Sb, Fe and Sr) were detected by using ICP-MS (NexION300X; Perkin Elmer, Waltham, MA, USA) at the Institute of Geochemistry, Chinese Academy of Sciences. Quality assurance and quality control were assessed by standard operating procedures, calibration with standards, and analysis of reagent blanks, with each batch of 20 water samples. Relative standard deviations for heavy metals were ~±5% and recovery percentage ranged from 90% to 110%. Otherwise, the samples were detected again until the data reach the standard.

## Statistical analysis

Multivariate statistical approaches, including correlation analysis and PCA, were used to identify the possible origin of nine heavy metals in the study area (*Ustaoğlu & Tepe, 2019*; *Xiao et al., 2019*). PCA is the most common approach to explore the sources of heavy metals by reducing the dimensionality of the dataset to several influencing factors (*Loska & Wiechuła, 2003*; *Zeng & Han, 2020*). The suitability of the datasets for the PCA method is assessed by Kaiser–Meyer–Olkin (KMO) value and Bartlett's sphericity test ($p < 0.001$) (*Varol, 2011*). Microsoft Office 2019 and the statistical software package SPSS 21.0 were used to perform all of the data processing.

## Health risk assessment

The hazard index (HI), the sum of both two pathway HQs, represents total potential non-carcinogenic risks for each heavy metal. If HQ or HI exceeds 1, it indicates a potentially adverse effect on human health and needs for further study (*Wang et al., 2017*). The HQ and HI are calculated as follows:

$$\text{ADD}_{\text{ingestion}} = (C_{\text{w}} \times \text{IR} \times \text{EF} \times \text{ED})/(\text{BW} \times \text{AT}) \tag{1}$$

$$\text{ADD}_{\text{dermal}} = \left(C_{\text{w}} \times \text{SA} \times K_p \times \text{ET} \times \text{EF} \times \text{ED} \times 10^{-3}\right)/(\text{BW} \times \text{AT}) \tag{2}$$

$$\text{HQ} = \text{ADD}/\text{RfD} \tag{3}$$

$$\text{RfD}_{\text{dermal}} = \text{RfD} \times \text{ABS}_{\text{GI}} \tag{4}$$

$$\text{HI} = \sum \text{HQ}s \tag{5}$$

where $ADD_{ingestion}$ and $ADD_{dermal}$ are the average daily doses via ingestion or dermal exposure (μg/kg/day), respectively. $C_w$ is the heavy metal concentration of each sample (μg/L); IR is the ingestion rate (L/day); EF is the exposure frequency (day/year); ED is the exposure duration (years); BW is the body weight (kg); AT is the average time (days); SA is the area of exposed skin (cm$^2$); $K_p$ is the dermal permeability coefficient for each heavy metal in water (cm/h); ET is the exposure time (h/day); $ABS_{GI}$ is the gastrointestinal absorption factor. The above parameters are from the United States Environmental Protection Agency (EPA) (*United States Environmental Protection Agency, 2004*).

## Water quality index

The WQI is calculated as follows:

$$WQI = \sum [W_i \times (C_i/S_i)] \times 100 \tag{6}$$

where $W_i$ is the weight of each element and represents different contributions to the overall water quality, which is calculated by the eigenvalues for each principal component (PC) and factor loading for each heavy metal from the PCA results. $C_i$ is the concentration of each heavy metal tested in this study. $S_i$ represents the limit value of drinking water for each heavy metal. According to the WQI values, water quality can be classified into five categories as excellent water (WQI < 50), good water (50 ≤ WQI < 100), poor water (100 ≤ WQI < 200), very poor water (200 ≤ WQI < 300) and undrinkable water (WQI ≥ 300). V and Sr are excluded from the WQI calculations due to the lack of official drinking water guidelines.

# RESULTS

## Kolmogorov–Smirnov test of data

Kolmogorov–Smirnov statistics indicate most of the water parameters and heavy metals were in non-normal distribution (Table 1). Therefore, we reported the median concentrations instead of mean concentrations in our study.

## Physicochemical characteristics

The physicochemical characteristics of water samples from the Lake Aha watershed are summarized in Table 1. Samples from three tributaries demonstrate slightly alkaline characteristics with a median pH value of 7.88, varying from 6.71 to 8.64. Relatively, the median pH value of the YR (7.64) is lower than the BR (8.05) and the JR (7.96). The DO values range from 0.42 to 18.64 mg/L, with a median of 9.59 mg/L. The EC values range from 199.4 to 1437.0 μS/cm, with a median of 584.8 μS/cm. Overall, the pH, DO and EC values in the Lake Aha watershed show a noticeable variability.

## Heavy metals content

The concentrations of dissolved heavy metals are also summarized in Table 1. According to their concentrations, these heavy metals can be divided into three groups: (1) Sr is in the highest level, with a median concentration of >100 μg/L; (2) Fe, Ba, Al, Ni and Sb are

**Table 1 Concentrations of dissolved heavy metals, pH, dissolved oxygen and electric conductivity in Lake Aha.**

| Tributaries | | pH | DO mg/L | EC µS/cm | Mn µg/L | Co µg/L | Al µg/L | Ni µg/L | Ba µg/L | V µg/L | Sb µg/L | Fe µg/L | Sr µg/L |
|---|---|---|---|---|---|---|---|---|---|---|---|---|---|
| Youyu River | Max | 8.62 | 18.64 | 1437.0 | 145.261 | 2.35 | 167.08 | 10.91 | 44.73 | 0.79 | 9.11 | 116.12 | 923.93 |
| | Min | 6.71 | 2.26 | 199.4 | 0.005 | 0.03 | 0.74 | 0.57 | 12.47 | 0.01 | 0.14 | 19.23 | 312.28 |
| | Median | 7.64 | 9.84 | 604.0 | 0.777 | 0.07 | 4.19 | 1.92 | 18.87 | 0.05 | 0.89 | 42.62 | 515.91 |
| | Mean | 7.62 | 9.32 | 737.2 | 16.458 | 0.25 | 17.17 | 2.68 | 20.96 | 0.16 | 1.06 | 48.48 | 539.36 |
| | SD[a] | 0.44 | 2.84 | 268.5 | 35.509 | 0.53 | 31.30 | 2.40 | 7.51 | 0.17 | 1.31 | 18.44 | 170.15 |
| | K–S test[b] | 0.200 | 0.000 | 0.000 | 0.000 | 0.000 | 0.000 | 0.000 | 0.000 | 0.000 | 0.000 | 0.000 | 0.076 |
| Baiyan River | Max | 8.42 | 10.85 | 1048.0 | 10.565 | 0.16 | 13.48 | 2.07 | 95.01 | 1.15 | 18.13 | 52.07 | 723.62 |
| | Min | 6.74 | 0.42 | 366.0 | 0.005 | 0.03 | 1.00 | 0.38 | 17.36 | 0.22 | 0.29 | 10.69 | 167.45 |
| | Median | 8.05 | 9.68 | 399.1 | 0.153 | 0.05 | 4.85 | 0.80 | 28.56 | 0.44 | 1.03 | 19.68 | 302.42 |
| | Mean | 7.89 | 8.18 | 501.9 | 0.754 | 0.07 | 5.01 | 0.99 | 35.84 | 0.55 | 2.73 | 23.63 | 360.58 |
| | SD | 0.48 | 3.66 | 227.5 | 1.877 | 0.04 | 2.70 | 0.46 | 22.64 | 0.28 | 3.81 | 11.97 | 165.07 |
| | K–S test | 0.000 | 0.000 | 0.000 | 0.000 | 0.000 | 0.108 | 0.000 | 0.000 | 0.002 | 0.000 | 0.000 | 0.000 |
| Jinzhong River | Max | 8.64 | 10.11 | 763.0 | 3.071 | 0.43 | 13.89 | 4.92 | 67.94 | 3.02 | 17.29 | 47.36 | 900.33 |
| | Min | 7.48 | 5.93 | 362.4 | 0.046 | 0.03 | 1.94 | 0.37 | 24.39 | 0.30 | 0.45 | 9.50 | 178.49 |
| | Median | 7.96 | 8.59 | 674.2 | 0.281 | 0.21 | 7.27 | 1.22 | 42.68 | 1.38 | 1.33 | 32.00 | 469.87 |
| | Mean | 7.94 | 8.14 | 594.8 | 0.570 | 0.18 | 7.39 | 1.47 | 41.93 | 1.29 | 2.64 | 28.80 | 557.77 |
| | SD | 0.27 | 1.22 | 148.37 | 0.85 | 0.12 | 3.78 | 1.14 | 9.72 | 0.78 | 4.00 | 10.30 | 229.47 |
| | K–S test | 0.200 | 0.054 | 0.000 | 0.000 | 0.172 | 0.145 | 0.088 | 0.089 | 0.200 | 0.000 | 0.193 | 0.062 |
| Total | Max | 8.64 | 18.64 | 1437.0 | 145.261 | 2.35 | 167.08 | 10.91 | 95.01 | 3.02 | 18.13 | 116.12 | 923.93 |
| | Min | 6.71 | 0.42 | 199.4 | 0.005 | 0.03 | 0.74 | 0.37 | 12.47 | 0.01 | 0.14 | 9.50 | 167.45 |
| | Median | 7.88 | 9.59 | 584.8 | 0.364 | 0.06 | 4.94 | 1.22 | 25.33 | 0.33 | 1.00 | 35.08 | 413.57 |
| | Mean | 7.76 | 8.75 | 642.7 | 8.650 | 0.18 | 11.57 | 1.93 | 29.39 | 0.48 | 1.87 | 37.08 | 485.30 |
| | SD | 0.45 | 2.97 | 261.7 | 26.352 | 0.39 | 22.97 | 1.94 | 16.80 | 0.56 | 2.96 | 19.22 | 198.11 |
| | K–S test | 0.000 | 0.000 | 0.000 | 0.000 | 0.000 | 0.000 | 0.000 | 0.000 | 0.000 | 0.000 | 0.002 | 0.000 |

**Notes:**
[a] Standard deviation.
[b] Kolmogorov–Smirnov test.

in intermediate levels, with median concentrations ranging from 1 to 100 µg/L; (3) Mn, V and Co are in low level, with median concentrations of <1 µg/L.

Table 2 shows the comparison of heavy metals concentrations between this study and the guideline values for drinking water set by *China Environmental Protection Administration (2006)*, *United States Environmental Protection Agency (2004)* and *World Health Organization (WHO) (2006)*. The guideline values set by *China Environmental Protection Administration (2006)* are the strictest and most comprehensive of the three. Thus, we used Chinese guideline values as the standard to evaluate water quality in this study. The median concentrations of nine heavy metals in this study are within limited values of drinking water guidelines (except V and Sr, without limited values in three guidelines), indicating the water is not heavily polluted by these heavy metals. However, for specific metals, their concentrations exceed the limit values at some sites (e.g., Mn at YR-5, Sb at BR-14). This indicates that these sites can attribute to a relatively high anthropogenic input. Besides, if the concentration of one sampling site is higher than other sampling sites (even if lower than

**Table 2 Comparison of heavy metals in three tributaries of Lake Aha with drinking water guidelines, worldwide river average and Xijiang River.**

| | Study area μg/L | Drinking water guidelines | | | World River[d] μg/L | Beipan River[e] μg/L | Nanpan River[e] μg/L |
|---|---|---|---|---|---|---|---|
| | | China[a] μg/L | USEPA[b] μg/L | WHO[c] μg/L | | | |
| Mn | 0.37 | 100 | | 400 | 7 | 0.30 | 0.30 |
| Co | 0.06 | 1,000 | | | 0.1 | 0.06 | 0.04 |
| Al | 4.94 | 200 | | 200 | 50 | 10.8 | 11.7 |
| Ni | 1.22 | 20 | | 70 | 0.3 | 0.34 | 0.27 |
| Ba | 25.33 | 700 | 2,000 | 700 | 20 | 25.4 | 25.1 |
| V | 0.33 | | | | 0.9 | 0.78 | 0.77 |
| Sb | 1.00 | 5 | | | 0.07 | 1.68 | 1.58 |
| Fe | 35.08 | 300 | | 300 | 40 | | |
| Sr | 413.57 | | | | 70 | 329 | 244 |

Notes:
[a] *China Environmental Protection Administration (2006).*
[b] *United States Environmental Protection Agency (EPA) (2003).*
[c] *World Health Organization (WHO) 2006.*
[d] *Li (1982).*
[e] *Liu et al. (2017).*

guideline values), such as the Al concentrations in YR-5, it may also reveal an impact of anthropogenic input.

## DISCUSSION

### Comparison and distribution of heavy metals in tributaries

In the low flow season, the temporal distribution of each heavy metals in tributaries is shown in Fig. 2. In general, the Al and Mn concentrations showed an increasing trend with time, whereas the Sb and Ba concentrations displayed a decreasing trend. Meanwhile, the trends of the same heavy metal between three tributaries could be different.
For example, the Co concentration of the JR increased with time in contrast to the stable trend in the other tributaries, and Ni concentrations of the YR and the JR was higher in November and December while BR did not change obviously. It is noteworthy that the JR showed more significant variation than the YR and BR, perhaps due to the small number of sampling sites ($n = 3$) on the JR.

The spatial distribution of heavy metals in three tributaries is shown in Fig. 3. Spatial distributions of heavy metals displayed an extensive variation, and we could conclude three primary patterns based on the spatial distribution features: (1) Al, Mn, Co and Ni, their concentrations increased in the middle reaches of the YR. (2) Fe and Sr, their concentrations were higher in YR but occurred valley values in the upstream. (3) Ba, V and Sb, their concentrations were consistently low in the YR but relatively high value in the other two tributaries.

Different anthropogenic activities may explain the distinction on dissolved heavy metals among three tributaries. The lowest pH value in the YR exhibited an influence of acid mine

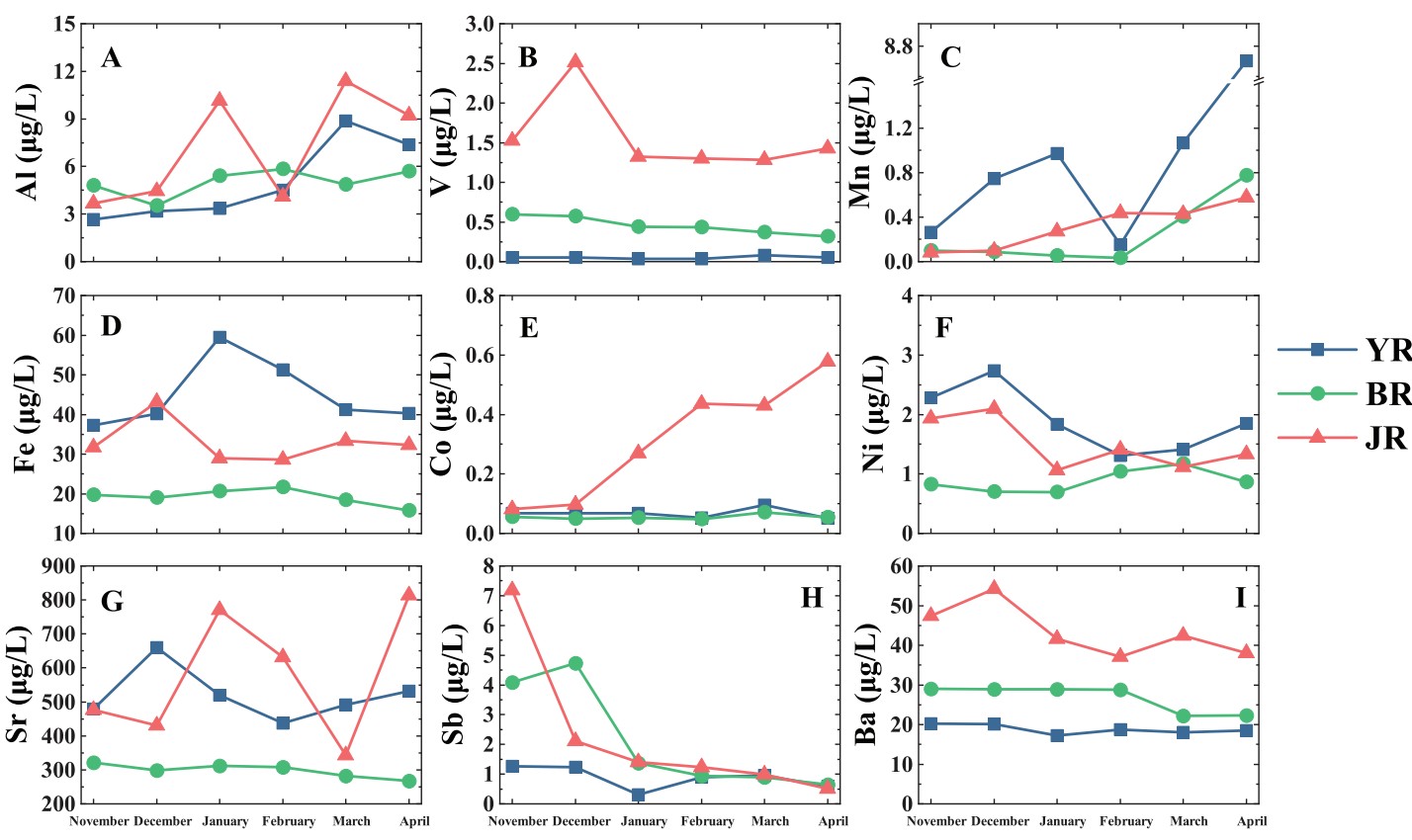

**Figure 2 Temporal distributions of dissolved heavy metals in three tributaries of Lake Aha.** YR, Youyu River; BR, Baiyan River; JR, Jinzhong River. The vertical dash line divides the different tributaries. (A) Al concentrations. (B) V concentrations. (C) Mn concentrations. (D) Fe concentrations. (E) Co concentrations. (F) Ni concentrations. (G) Sr concentrations. (H) Sb concentrations. (I) Ba concentrations.

drainage. The JR was the most typical urban tributary impacted by human activities in our study, and the Jinyang Wastewater Treatment Plant with limited capacity and damaged sewer networks provided an amount of pollutant input (*Huang et al., 2009*; *Song et al., 2011*; *Zhang et al., 2019*). The peak value in BR-14 (a groundwater well) could attribute to the contamination of domestic sewage through karst conduits.

We compare our results with previous results of the upstream of Xijiang River (Beipan River and Nanpan River), Guizhou, China and the world average concentrations (*Li, 1982*; *Liu et al., 2017*) (Table 2). The Ni, Sb and Sr concentrations of our study were higher than the worldwide river average concentrations, while Co and Fe concentrations were comparable. Mn, Co, Ba, V and Sb concentrations were comparable to that in the upstream of Xijiang River, but the Sr concentration in both sites was much higher than the worldwide river average. Notably, the geological background of the upstream of Xijiang River was consistent with our study site, indicating that the bedrock lithology, as a way of natural source, could significantly affect dissolved heavy metal contents in water, and the concentrations of Sr exposed to carbonate rocks might be higher than that in other regions (*Han & Liu, 2006*; *Han et al., 2019*; *Peng et al., 2012*; *Zeng et al., 2020*).

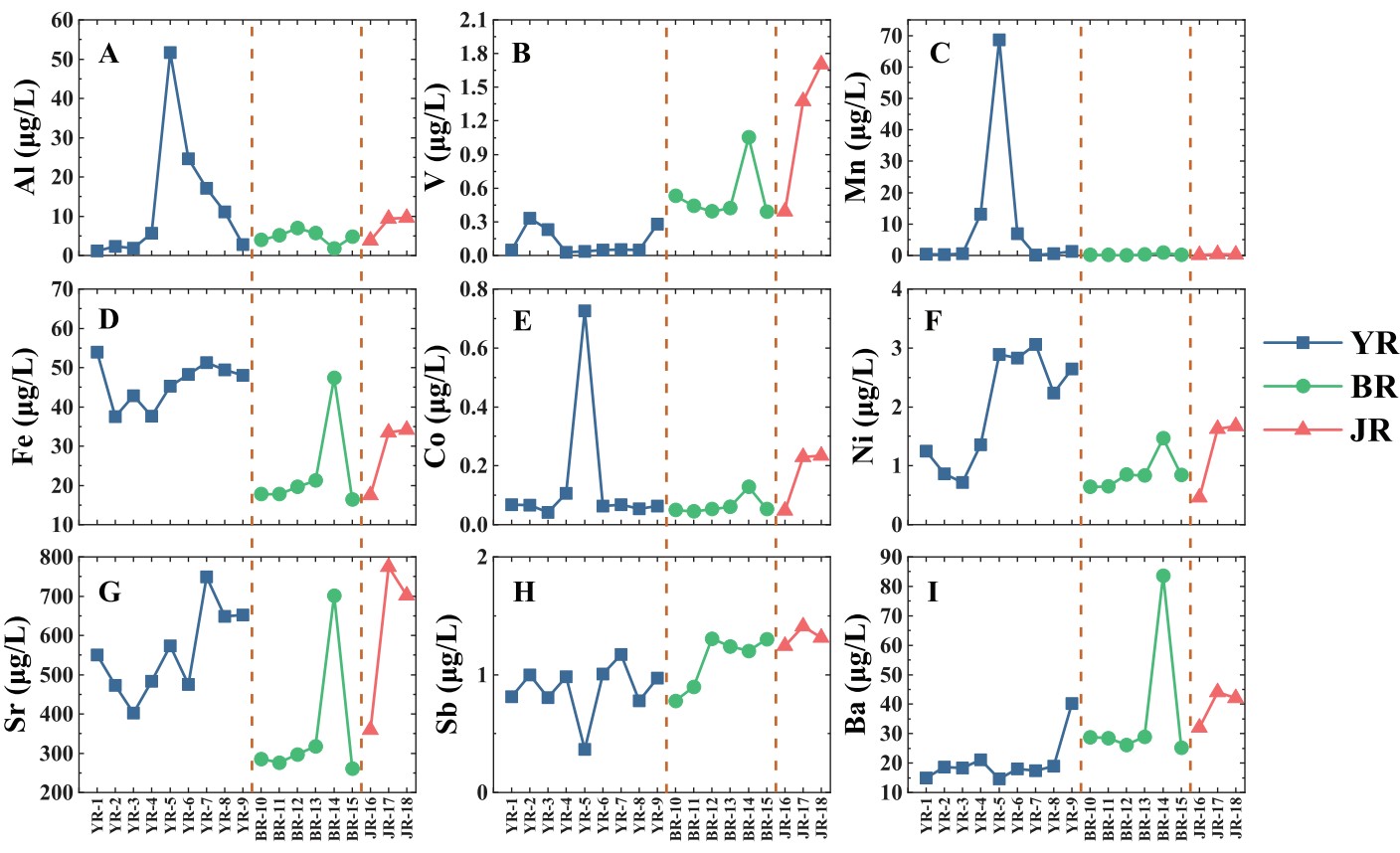

**Figure 3 Spatial distributions of dissolved heavy metals in three tributaries of Lake Aha.** YR, Youyu River; BR, Baiyan River; JR, Jinzhong River. The vertical dash line divides the different tributaries. (A) Al concentrations. (B) V concentrations. (C) Mn concentrations. (D) Fe concentrations. (E) Co concentrations. (F) Ni concentrations. (G) Sr concentrations. (H) Sb concentrations. (I) Ba concentrations.

## Potential source identification

### Correlation analysis

We used Spearman's correlation to explore the associations and interactions of nine dissolved heavy metals and physicochemical characteristics in three tributaries. The results are presented in Table 3. The pH is an important factor affecting the dissolved heavy metal contents in water. The lower pH can increase the competition between metals and hydrogen ions for binding sites and may dissolve metal-carbonate complexes in sediments, releasing free metals (*Papafilippaki, Kotti & Stavroulakis, 2008*). As shown in Table 3, pH was significant positive correlated ($p < 0.01$) with Al (0.383), V (0.247), Sb (0.339), while Mn (−0.299), Fe (−0.399) and Sr (−0.303) were significant negative correlated ($p < 0.01$) with pH. The results indicated that pH played a crucial role for the dissolved heavy metals in three tributaries (except Co, Ni and Ba). In addition to pH, the source is considered to be the most vital factor in the temporal-spatial distribution of heavy metals in water, which is integrated by a natural source and anthropogenic source (*Li & Zhang, 2010*; *Liu et al., 2017*). EC, as a measure of total dissolved ions in water, is largely a function of basin biogeochemistry and land use. Increases in EC might be accompanied by elevated dissolved heavy metals in water (*Walker & Pan, 2006*).

**Table 3 Pearson correlation matrix of heavy metals and physicochemical parameters in the tributaries of Lake Aha.**

|     | Mn | Co | Al | Ni | Ba | V | Sb | Fe | Sr | pH | DO | EC |
|-----|------|------|------|------|------|------|------|------|------|------|------|------|
| Mn  | 1 | | | | | | | | | | | |
| Co  | 0.536** | 1 | | | | | | | | | | |
| Al  | 0.198* | 0.239* | 1 | | | | | | | | | |
| Ni  | 0.455** | 0.490** | 0.282** | 1 | | | | | | | | |
| Ba  | −0.209* | 0.199* | −0.214* | −0.078 | 1 | | | | | | | |
| V   | −0.289** | 0.210* | −0.231* | −0.284** | 0.719** | 1 | | | | | | |
| Sb  | −0.406** | 0.063 | −0.14 | −0.107 | 0.424** | 0.437** | 1 | | | | | |
| Fe  | 0.337** | 0.291** | 0.024 | 0.641** | −0.174 | −0.408** | −0.193 | 1 | | | | |
| Sr  | 0.274** | 0.377** | 0 | 0.487** | 0.05 | 0.015 | 0.016 | 0.602** | 1 | | | |
| pH  | −0.299** | −0.059 | 0.383** | −0.083 | 0.083 | 0.247* | 0.339** | −0.399** | −0.303** | 1 | | |
| DO  | −0.02 | −0.062 | 0.383** | 0.035 | −0.435** | −0.292** | −0.079 | −0.105 | −0.157 | 0.443** | 1 | |
| EC  | 0.295** | 0.336** | −0.147 | 0.552** | 0.064 | −0.026 | 0.047 | 0.633** | 0.679** | −0.310** | −0.239* | 1 |

Notes:
* Strong positive correlation coefficients at the 0.05 level (two-tailed).
** Strong positive correlation coefficients at the 0.01 level (two-tailed).

Therefore, distinguishing the effects of natural and anthropogenic sources of variability in conductivity is important for identifying the potential sources for dissolved heavy metals. EC was strong positive correlation ($p < 0.01$) with Mn (0.295), Co (0.336), Ni (0.552), Fe (0.633) and Sr (0.679) in our study. For dissolved heavy metals, strong positive correlations indicated that different heavy metals might have similar sources, migration and conversion behaviors (*Wang et al., 2017*). Strong positive correlations ($p < 0.01$) were observed ranging from 0.274 to 0.602 between each pair of Mn, Co, Ni, Fe and Sr. Co was positively correlated with most elements (except Sb), indicating that Co could have multiple sources in the Lake Aha watershed. Moreover, Ba, V and Sb were observed of pairwise strong correlations ($p < 0.01$), indicating that they had similar sources.

## PC analysis

Principal component analysis is applied for nine heavy metals in three tributaries to identify their possible sources. The KMO value and Bartlett's sphericity test are respectively 0.69 and zero, indicating that the data in this study is suitable for PCA. Three PCs extracted from the element were shown in Table 4 and Fig. 4, accounting for 74.74% of the total variance.

The PC 1 explained 34.58% of the total variance and had a high loading of Mn (0.96), Co (0.95), Al (0.79) and Ni (0.74). The concentrations of these four heavy metals occurred an increase in the middle reaches of YR, where most of the abandoned coal mining was located (*Bai et al., 2007*; *Bangjiang et al., 2014*; *Pan et al., 2019*). Hence, the PC 1 with high loadings on Mn, Co, Al and Ni might be dominantly influenced by acid coal mine drainage (*Abraham, Dowling & Florentine, 2018*; *Ali et al., 2017*; *Santelli et al., 2010*; *Seo et al., 2017*; *Tiwary, 2001*). The PC 2, accounting for 22.61% of the total variance and predominantly contained Ba (0.88), V (0.84) and Sb (0.64), where the concentrations matched well with the upper Xijiang River (*Liu et al., 2017*). Among three tributaries, Ba, V

Table 4 Varimax rotated component matrix of dissolved heavy metals in the tributaries of Lake Aha.

| Variables | PC1 | PC2 | PC3 | Communalities |
|---|---|---|---|---|
| Mn | 0.96 | −0.12 | 0.07 | 0.93 |
| Co | 0.95 | 0.06 | 0.05 | 0.92 |
| Al | 0.79 | −0.20 | 0.11 | 0.67 |
| Ni | 0.74 | −0.05 | 0.43 | 0.72 |
| Ba | −0.08 | 0.88 | 0.16 | 0.75 |
| V | −0.11 | 0.84 | 0.06 | 0.79 |
| Sb | −0.03 | 0.64 | −0.10 | 0.73 |
| Sr | 0.02 | 0.25 | 0.85 | 0.42 |
| Fe | 0.33 | −0.18 | 0.78 | 0.80 |
| Eigenvalues (%) | 3.11 | 2.03 | 1.58 | |
| Variance (%) | 34.58 | 22.61 | 17.55 | |
| Cumulative (%) | 34.58 | 57.19 | 74.74 | |

**Notes:**
Extraction method: principal component analysis.
Rotation method: varimax with Kaiser normalization.

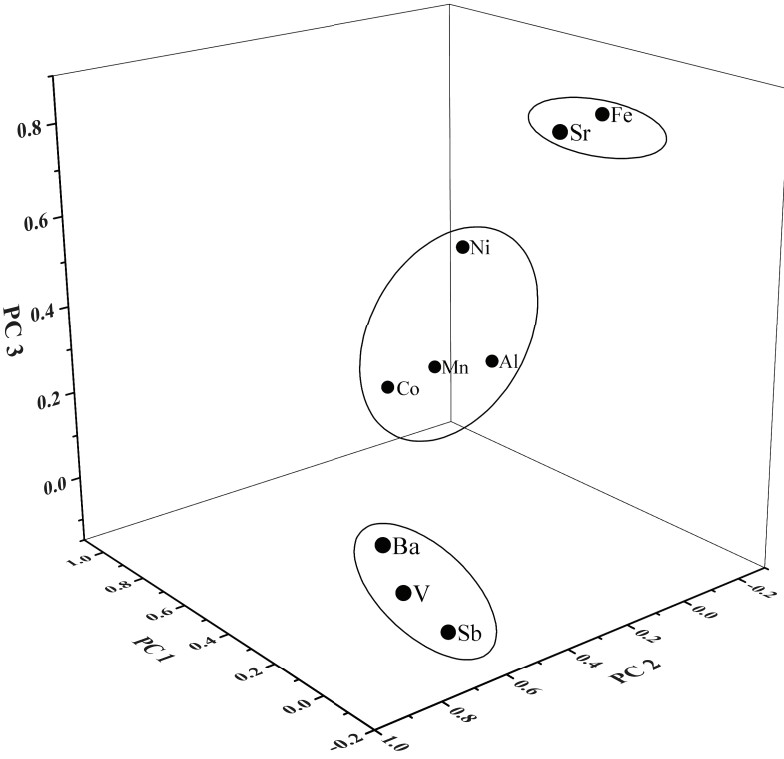

**Figure 4 Loading plot of factors or dissolved heavy metals in the tributaries of Lake Aha.**

and Sb show higher concentrations in BR and JR than YR. Previous investigation and the scene observation found that BR and JR are influenced by human activities more than YR, with a higher percentage of residential and commercial land (*Huang et al., 2009*; *Song et al.,*

**Table 5 Hazard quotient and hazard index for each heavy metal in the tributaries of Lake Aha.**

| PC | Eigenvalues | Relative eigenvalue | Variable | Loading value | Relative loading value on same PC | Weight |
|----|-------------|---------------------|----------|---------------|------------------------------------|--------|
| 1 | 3.11 | 0.46 | Mn | 0.96 | 0.25 | 0.12 |
| | | | Co | 0.95 | 0.25 | 0.12 |
| | | | Al | 0.79 | 0.21 | 0.10 |
| | | | Ni | 0.74 | 0.20 | 0.09 |
| | | | Fe | 0.33 | 0.09 | 0.04 |
| | | | Total | 3.76 | 1 | 0.46 |
| 2 | 2.03 | 0.30 | Ba | 0.88 | 0.37 | 0.11 |
| | | | V | 0.84 | 0.36 | 0.11 |
| | | | Sb | 0.64 | 0.27 | 0.08 |
| | | | Total | 2.36 | 1 | 0.30 |
| 3 | 1.58 | 0.24 | Sr | 0.89 | 0.42 | 0.10 |
| | | | Fe | 0.82 | 0.38 | 0.09 |
| | | | Ni | 0.43 | 0.20 | 0.05 |
| | | | Total | 2.14 | 1 | 0.23 |
| Total | 6.73 | | | | | 1 |

**Note:**
Weight was calculated by relative eigenvalue times relative loading value.

*2011*; *Zhang et al., 2019*). Considering that Ba, V and Sb are widely used in industrial and medical (*Filella, Belzile & Chen, 2002*; *Hope, 2008*; *Kravchenko et al., 2014*; *Wilson et al., 2010*), these heavy metals could be attributed to anthropogenic sources. The PC 3 had strong loadings of Fe (0.85) and Sr (0.78), with a variance of 17.55%. Fe is a major element in Earth's crust (*Wang et al., 2017*), therefore the high concentrations of Sr (much higher than the world average and consistent with the background value) may be originated from carbonate weathering (*Liu et al., 2017*). Therefore, PC 3 could be attributed to the sources of geologic origins.

## WQI and health risk assessment

According to PCA, the weights of each heavy metal are shown in Table 5. The WQI values of each sampling site were calculated with Eq. (6) and shown in Fig. 5. The WQI values varied from 3.12 to 15.64, with an average of 5.79. YR-5 (WQI = 15.64) was the only site of which the WQI value exceeded 10, regarded as a relatively high site. Therefore, all the water samples could be defined as excellent water quality (WQI < 50), indicating that the natural water in tributaries was suitable for drinking with respect to heavy metals pollution.

According to Eqs. (2)–(6), the HQ and HI values of heavy metals for adults and children by ingestion and dermal pathways are calculated and summarized in Table 6. All $HQ_{ingestion}$, $HQ_{dermal}$ and HI values are far smaller than 1, indicating these dissolved heavy metals in the Lake Aha watershed only make a little hazard for adults and children via ingestion and dermal absorption. Compared to dermal, ingestion is the primary pathway to endanger humans for these dissolved heavy metals. The HI value of Sb for children (0.154) is the only value exceeding 0.1. Consequently, Sb might be a potential

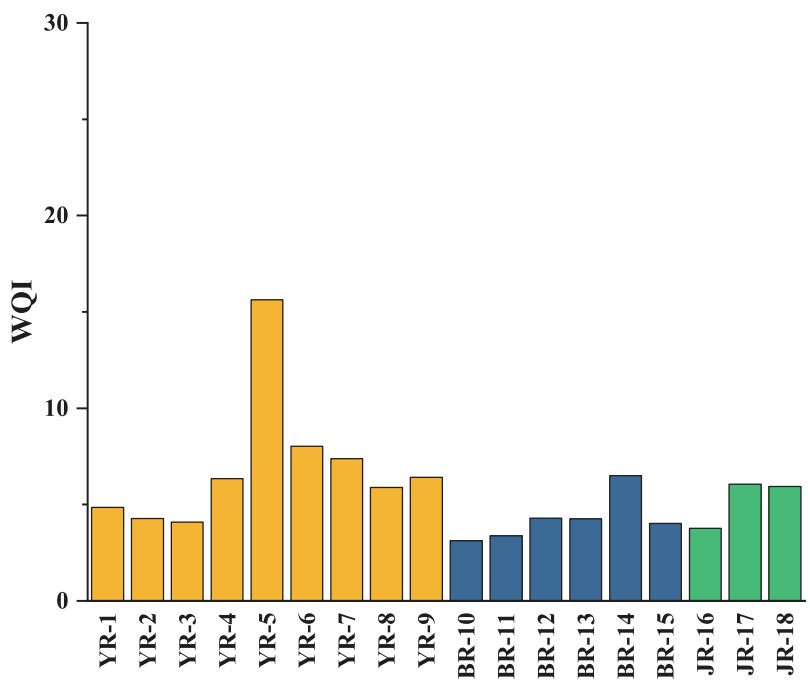

**Figure 5  WQI values of water in three tributaries of Lake Aha.**

**Table 6  Hazard quotient (HQ) and reference dose (RfD) for dissolved heavy metals in three tributaries of Lake Aha.**

| | $HQ_{ingestion}$ | | $HQ_{dermal}$ | | HI | | $K_p^a$ | $RfD_{ingestion}^{b,c}$ | $RfD_{dermal}^{b,c}$ |
|---|---|---|---|---|---|---|---|---|---|
| | Adults | Children | Adults | Children | Adults | Children | | | |
| Mn | $4.27 \times 10^{-4}$ | $6.37 \times 10^{-4}$ | $5.57 \times 10^{-5}$ | $1.64 \times 10^{-4}$ | $4.82 \times 10^{-4}$ | $8.01 \times 10^{-4}$ | $1.00 \times 10^{-3}$ | 24 | 0.96 |
| Co | $5.66 \times 10^{-3}$ | $8.45 \times 10^{-3}$ | $5.90 \times 10^{-5}$ | $1.74 \times 10^{-4}$ | $5.71 \times 10^{-3}$ | $8.62 \times 10^{-3}$ | $4.00 \times 10^{-4}$ | 0.3 | 0.06 |
| Al | $1.35 \times 10^{-4}$ | $2.02 \times 10^{-4}$ | $3.53 \times 10^{-6}$ | $1.04 \times 10^{-5}$ | $1.39 \times 10^{-4}$ | $2.13 \times 10^{-4}$ | $1.00 \times 10^{-3}$ | 1,000 | 200 |
| Ni | $1.67 \times 10^{-3}$ | $2.50 \times 10^{-3}$ | $4.36 \times 10^{-5}$ | $1.29 \times 10^{-4}$ | $1.71 \times 10^{-3}$ | $2.62 \times 10^{-3}$ | $2.00 \times 10^{-4}$ | 20 | 0.8 |
| Ba | $3.47 \times 10^{-3}$ | $5.18 \times 10^{-3}$ | $2.59 \times 10^{-4}$ | $7.63 \times 10^{-4}$ | $3.73 \times 10^{-3}$ | $5.95 \times 10^{-3}$ | $1.00 \times 10^{-3}$ | 200 | 14 |
| V | $9.05 \times 10^{-3}$ | $1.35 \times 10^{-2}$ | $4.73 \times 10^{-3}$ | $1.39 \times 10^{-2}$ | $1.38 \times 10^{-2}$ | $2.75 \times 10^{-2}$ | $1.00 \times 10^{-3}$ | 1 | 0.01 |
| Sb | $6.82 \times 10^{-2}$ | $1.02 \times 10^{-1}$ | $1.78 \times 10^{-2}$ | $5.25 \times 10^{-2}$ | $8.60 \times 10^{-2}$ | $1.54 \times 10^{-1}$ | $1.00 \times 10^{-3}$ | 0.4 | 0.008 |
| Fe | $1.37 \times 10^{-3}$ | $2.05 \times 10^{-3}$ | $3.58 \times 10^{-5}$ | $1.06 \times 10^{-4}$ | $1.41 \times 10^{-3}$ | $2.16 \times 10^{-3}$ | $1.00 \times 10^{-3}$ | 700 | 140 |
| Sr | $1.89 \times 10^{-2}$ | $2.82 \times 10^{-2}$ | $4.93 \times 10^{-4}$ | $1.45 \times 10^{-3}$ | $1.94 \times 10^{-2}$ | $2.97 \times 10^{-2}$ | $1.00 \times 10^{-3}$ | 600 | 120 |

Notes:
[a] *United States Environmental Protection Agency (2004)*.
[b] *Wang et al. (2017)*.
[c] *Wu et al. (2009)*.

non-carcinogenic risk to human health. It is worth noting that the HQ and HI values for children are higher than that for adults, suggesting that children are at higher risk than adults under the exposure of heavy metals.

## CONCLUSIONS

The heavy metal concentrations in the Lake Aha watershed displayed both temporal and spatial variations. The JR showed more obvious temporal variation than the YR and the

BR, and nine heavy metals could be classified into three spatial distribution patterns due to their features: (1) Al, Mn, Co and Ni; (2) Fe and Sr; (3) Ba, V and Sb. Sr was the most abundant heavy metals and the only element of which concentration exceeding 100 µg/L. The PCA results indicated that the acid coal mine drainage might influence Mn, Co, Al and Ni concentrations, and Ba, V and Sb could be attributed to anthropogenic activities of industrial and medical, and Fe and Sr mainly presented a natural geological feature in the study area. The heavy metal concentrations in most sampling sites were within the limited values for the Chinese drinking water guideline, and the water quality indicated that the waters in three tributaries were not polluted by the nine heavy metals. Overall, the water quality of the Lake Aha watershed is good, however risk assessment suggested that Sb had a relatively higher HI, suggesting a potential risk. Children were under a higher risk than adults, while the ingestion was the primary exposure pathway. To explore the behaviors of heavy metals during water/particle interaction, there is a need for further research including the behavior in suspended particulate matter and the heavy metals speciation analysis in future research.

## ACKNOWLEDGEMENTS

We thank Huipeng Jia and Yuanyi Shen from Guizhou University for sample analyses and Prof. Runsheng Yin from Institute of Geochemistry, Chinese Academy of Sciences for revising work.

### Funding

This research was funded by the National Natural Science Foundation of China (41863004, 41863003, 41763019), the Joint Fund of the National Natural Science Foundation of China and Guizhou Province, China (U1612442), the first-class discipline construction project in Guizhou Province—Public Health and Preventive Medicine (No. 2017[85], GNYL [2017] 007), and the Guizhou Science and Technology Support Program ([2019]2832, [2016]1028). The funders had no role in study design, data collection and analysis, decision to publish, or preparation of the manuscript.

### Grant Disclosures

The following grant information was disclosed by the authors:
National Natural Science Foundation of China: 41863004, 41863003 and 41763019.
National Natural Science Foundation of China and Guizhou Province, China: U1612442.
Guizhou Province—Public Health and Preventive Medicine: 2017 [85], GNYL [2017] 007.
Guizhou Science and Technology Support Program: [2019]2832 and [2016]1028.

### Competing Interests

The authors declare that they have no competing interests.

## Author Contributions

- Shilin Gao conceived and designed the experiments, performed the experiments, analyzed the data, prepared figures and/or tables, authored or reviewed drafts of the paper, and approved the final draft.
- Zhuhong Wang conceived and designed the experiments, authored or reviewed drafts of the paper, and approved the final draft.
- Qixin Wu conceived and designed the experiments, authored or reviewed drafts of the paper, and approved the final draft.
- Jie Zeng analyzed the data, authored or reviewed drafts of the paper, and approved the final draft.

## Data Availability

Raw data is available in Figs. 2–4 and the Supplemental Files.

## Supplemental Information

Supplemental information for this article can be found online at http://dx.doi.org/10.7717/peerj.9660#supplemental-information.

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
