# Peer review of "Multivariate statistical evaluation of dissolved heavy metals and a water quality assessment in the Lake Aha watershed, Southwest China"

_PeerJ, doi:10.7717/peerj.9660_

## Round 0.1 · original submission · Major Revisions

While the reviewers indicated acceptance with minor revisions, my reading of their comments suggests more than a minor revision. Please make sure that you address the reviewers suggestions in detail.

·

Basic reporting

1. The manuscript presents assessment of dissolved heavy metals and a water quality assessment in the Lake Aha watershed, Southwest China with a combined approach using statistical analysis and water quality index, which is interesting. The subject addressed is within the scope of the journal.
2. However, the manuscript, in its present form, contains several weaknesses. Appropriate revisions to the following points should be undertaken in order to justify recommendation for publication.

Experimental design

3. For readers to quickly catch your contribution, it would be better to highlight major difficulties and challenges, and your original achievements to overcome them, in a clearer way in abstract and introduction.
4. It is mentioned in line 71 that Lake Aha watershed is adopted as the case study. What are other feasible alternatives? What are the advantages of adopting this particular case study over others in this case? How will this affect the results? The authors should provide more details on this.
5. It is mentioned in line 143 that the standard of Chinese drinking water are adopted as benchmark for comparison. What are the other feasible alternatives? What are the advantages of adopting these particular standards over others in this case? How will this affect the results? More details should be furnished.
6. It is mentioned in line 106 that multivariate statistical methods are adopted to assist the interpretation of the relationships between the parameters and the results. What are the other feasible alternatives? What are the advantages of adopting these particular methods over others in this case? How will this affect the results? The authors should provide more details on this.
7. It is mentioned in line135 that Water Quality Index is adopted to evaluate water quality status. What are the other feasible alternatives? What are the advantages of adopting this particular index over others in this case? How will this affect the results? The authors should provide more details on this.
8. It is mentioned in line 117 that hazard quotient and hazard index are adopted to evaluate health risk status. What are the other feasible alternatives? What are the advantages of adopting these particular indices over others in this case? How will this affect the results? The authors should provide more details on this.
9. It is mentioned in line96 that several physicochemical parameters are adopted in this study. What are the other feasible alternatives? What are the advantages of adopting these particular parameters over others in this case? How will this affect the results? The authors should provide more details on this.
10. It is mentioned in line 107 that PCA were used to identify the possible origin of nine heavy metals in the study area. What are the other feasible alternatives? What are the advantages of adopting this particular technique over others in this case? How will this affect the results? The authors should provide more details on this.
11. What statistical method is adopted to analyze spatial and temporal changes in water quality. The authors need to present statistical analyses in the text.
12. It is mentioned in line 156 that "… Overall, the pH, DO and EC values in the Lake Aha watershed show a noticeable variability. .…" More justification should be furnished on this issue. how and when the spatial and temporal changes in water quality occur?

Validity of the findings

13. It is mentioned in line 195 that "…We compare our water samples with the data of other rivers (Li et al. 2007). Mn, Co, Ba and V, heavy metal concentrations of our study were higher than world average concentrations but were significantly lower than the Danjiangkou Reservoir …" More justification should be furnished on this issue. The author may present a table to show comparisons.
14. Some key parameters are not mentioned. The rationale on the choice of the particular set of parameters should be explained with more details. Have the authors experimented with other sets of values? What are the sensitivities of these parameters on the results?
15. Some assumptions are stated in various sections. Justifications should be provided on these assumptions. Evaluation on how they will affect the results should be made.
16. The discussion section in the present form is relatively weak and should be strengthened with more details and justifications.

Additional comments

17. Moreover, the manuscript could be substantially improved by relying and citing more on recent literatures about contemporary real-life case studies on sustainability and/or water quality such as the followings:

Ustaoğlu, F., Tepe, Y., & Taş, B. (2019). Assessment of stream quality and health risk in a subtropical Turkey river system: A combined approach using statistical analysis and water quality index. Ecological Indicators, 105815.

Ustaoğlu Fikret, Tepe Yalçın, Aydin Handan (2020) Heavy metals in Sediments of Two Nearby Streams from Southeastern Black Sea Coast: Contamination and Ecological Risk Assessment
Environmental Forensics 21 (02), 145-156

Ustaoglu, F., Tepe, Y., Aydin, H., & Akbas, A. (2020). EVALUATION OF SURFACE WATER QUALITY BY MULTIVARIATE STATISTICAL ANALYSES AND WQI: CASE OF COMLEKCI STREAM,(GIRESUN-TURKEY). FRESENIUS ENVIRONMENTAL BULLETIN, 29(1), 167-177.

Taş, B., Tepe, Y., Ustaoğlu, F., & Alptekin, S. (2019). Benthic algal diversity and water quality evaluation by biological approach of Turnasuyu Creek, NE Turkey. Desalination and water treatment, 155, 402-415.

Ustaoğlu, F., & Tepe, Y. (2019). Water quality and sediment contamination assessment of Pazarsuyu Stream, Turkey using multivariate statistical methods and pollution indicators. International Soil and Water Conservation Research, 7(1), 47-56.

18. In the conclusion section, the limitations of this study, suggested improvements of this work and future directions should be highlighted.

·

Basic reporting

This manuscript presents new data on the heavy metals at 18 sampling points with 108 samples in the Lake Aha watershed, Southwest China. The authors interpret their heavy metals results by applying a couple of statistical techniques to better understand their data along with distribution, sources, controlling factors, and water quality for human consumption. Although the analysis and methods used are nothing new, I assume that this kind of study will continue in the future and this first compilation will be essential. As the quality of surface water is one of the most sensitive issues worldwide, this study will be of great importance not only at a local scale but all over the world and the amount of data on a new river system is worthy of publication. However, there are some major and minor issues needed to be considered prior to publication.

Experimental design

In fact, the paper mostly deals with the heavy metals with some minor description of the pH and even minor of the EC and DO from abstract to the conclusion. I recommend a more appropriate description of the physicochemical variables along with the heavy metals.

Validity of the findings

In statistics, the consistency of data is fundamental: if the dataset is not adequate, the output of the statistical analysis is not significant. For example, are there any missing data? Please do clarify. In the same section, before applying Pearson’s correlation, did you test the normality of the dataset? I do not think the dataset is normal in this study. If data are not normal you have to use Spearman’s correlation. Please, make sure and elaborate it.

Additional comments

The authors did not acknowledge the importance of pH and EC for the abundance and distribution of heavy metals in the inflowing tributaries. Specify that these issues are within the scope of this study.
In addition, maybe authors should explain better that on what basis they select the sampling points. Why only three sampling points from the Jinzhong River basin? It is not sufficient to show a map! In addition to how many points are selected, why you select these points is more important. In general, a higher description of the hydrology and climatology of the basins must be added

---

## Round 0.2 · accepted · Accept

I commend you for making the effort to address the reviewers comments so thoroughly. The paper is much better because of your attention to the reviewers comments.